# The Kocher–Caird Criteria for Pediatric Septic Arthritis of the Hip: Time for a Change in the *Kingella* Era?

**DOI:** 10.3390/microorganisms12030550

**Published:** 2024-03-10

**Authors:** Silvia Valisena, Giacomo De Marco, Oscar Vazquez, Blaise Cochard, Christina Steiger, Romain Dayer, Dimitri Ceroni

**Affiliations:** 1Pediatric Orthopedic Unit, Pediatric Surgery Service, Geneva University Hospitals, 1205 Geneva, Switzerland; 2Division of Orthopedic and Trauma Surgery, Geneva University Hospitals, 1205 Geneva, Switzerland

**Keywords:** pediatric osteoarticular infection, septic arthritis, *Kingella kingae*, prediction rule, algorithm

## Abstract

Pediatric septic arthritis of the hip (SAH) in children is a severe pathology, requiring prompt diagnosis and treatment to avoid destructive sequelae of the joint. Its diagnosis can be challenging, however, due to its spectrum of manifestations and differential diagnosis. Last century, multiple research teams studied the curves of systemic inflammation markers to aid the differential diagnosis. Kocher showed that a history of fever >38.5 °C, non-weight bearing, an erythrocyte sedimentation rate >40 mm/h, and serum white blood cells >12,000/mm^3^ were highly suggestive of SAH, with a predicted probability of 99.6% when all these predictors manifested in pediatric patients. Caird validated these criteria, also adding a C-reactive protein >20 mg/L, reaching a 98% probability of SAH when these five criteria were present. The Kocher and the Caird criteria were then applied in multiple settings, but were never clearly validated. Moreover, they were studied and validated in the years when *Kingella kingae* was just emerging, and this was probably responsible for false-negative cases in multiple centers. For this reason, the Kocher and the Caird criteria are still at the center of a debate on the diagnostic tools for pediatric SAH. We provide a historical overview of the development of clinical and laboratory test algorithms for pediatric SAH. Further, new perspectives for future research on the prediction rules of pediatric SAH are here proposed.

## 1. Septic Arthritis of the Hip in Children

A huge body of literature in the last century on pediatric osteoarticular infections is dedicated to septic arthritis of the hip (SAH). This joint infection affects children of all ages and is a medical and surgical emergency, requiring prompt diagnosis and treatment. The overall incidence of pediatric septic arthritis, irrespective of the joint involved, is 1–5 per 100,000 children in Western countries [1,2,3]. However, with regard to the hip, there are no robust data on its incidence in children. Possible reasons for this include the challenges of diagnosing SAH and its differential diagnosis. The debate has lasted decades and still has no definitive answer.

Pediatric SAH presents a spectrum of clinical signs and symptoms. These range from pain in the affected hip, limping or a refusal to bear weight in children of walking age, and contracture in flexion, abduction, and external rotation of the hip due to the joint capsule’s effusion-related distention. These symptoms occur in a debilitated child with systemic symptoms like fever, dehydration, loss of appetite, and even lethargy [4]. 

The challenge lies in the fact that the combination of these signs and symptoms is not specific to SAH. They can also manifest in multiple other infectious and non-infectious hip conditions in neonates, children, and adolescents. Among the main etiologies to consider in these cases are traumas (including child abuse), inflammatory diseases like transient synovitis of the hip (TSH), the spectrum of juvenile idiopathic arthritis, rheumatic fever, and Legg–Calvé–Perthes disease. Less common etiologies are sickle cell anemia, slipped capital femoral epiphysiolysis, malignancies like leukemia, villonodular synovitis, and soft tissue sarcomas, but other infectious diseases can also be mentioned, such as an accompanying or isolated osteomyelitis, psoas abscesses, or pyogenic sacroiliitis [4,5]. Although many of these pediatric hip conditions tend to affect specific age ranges, SAH can manifest itself at any age, with either an insidious or sudden onset, thus presenting a challenge for differential diagnosis. 

## 2. The Diagnosis and Differential Diagnosis of Pediatric Septic Arthritis of the Hip

SAH can lead to the development of destructive pathological changes, especially when pyogenic pathogens are involved. Bacterial enzymes activate the process of chondrolysis within 6–12 h, which can progress to joint destruction within 5–7 days [6]. Thus, a prompt diagnosis aided by blood cultures and, above all, an examination of joint fluid aspirate (JFA) is essential; subsequent treatment via needle aspiration lavage, arthrotomy, or even arthroscopy with postoperative antibiotics is fundamental [7,8].

In the context of such invasive diagnostic procedures and treatment required for prompt management of SAH, a differential diagnosis with the ability to exclude SAH or confirm a milder pathology would be very welcome. That other diagnosis is often TSH, which is probably one of the most common pediatric hip conditions. TSH and SAH share similar symptoms and signs, including problematic hip joint effusion in a wide age range of the pediatric population, from 3 to 10 years old (mean age: 4.7) for TSH and with no age limitations for SAH [9]. The disorders are differentiated by interpreting JFA analyses and blood cultures [10]. From the start of the 20th century, JFA analysis was the primary method used to demonstrate the presence of pathogens using Gram staining and JFA cultures accompanied by blood cultures to ascertain the septic origin of the hip joint effusion according to Koch’s criteria [10,11,12]. 

Until the end of the 20th century, there was no consensus on the JFA white blood cell (WBC) count threshold for determining SAH, which varied from 20,000 to 50,000/mm^3^. Diagnoses relied instead on the JFA’s appearance, Gram stains and cultures, and blood cultures [13,14,15,16]. However, clinical reports began to show that, based on the JFA and blood cultures, SAH could be diagnosed when (1) the JFA WBC count is > 50,000/mm^3^, JFA Gram staining and cultures are positive, and blood cultures are positive; (2) the JFA WBC count is >50,000/mm^3^, with either positive JFA Gram staining and cultures or positive blood cultures; or (3) the JFA WBC count is >50,000/mm^3^, with negative JFA Gram staining and cultures and negative blood cultures, in so-called ‘culture-negative’ cases [11,15].

## 3. The Development of Prediction Rules for Pediatric Septic Arthritis of the Hip

It was in this context that a search began for biochemical serum markers that would provide unequivocal parameters and a certain differential diagnosis of either SAH or TSH before any invasive procedure (studies shown in Table 1). The Kocher criteria found their roots in multiple studies, conducted from the 1970s onward, investigating the diagnostic role of serum WBC count and differentiation and of the erythrocyte sedimentation rate (ESR) in cases of TSH and SAH [13,15,17,18]. It is of note that Morrey et al. (1975) were the first to observe trends in the ESR and WBC count and differentiation in children with SAH [17]. Kunnamo et al. (1987) were the first to propose predictive criteria for distinguishing between septic arthritis (of any joint) and juvenile arthritis in children using univariate and multivariate logistic regression analyses based on clinical predictors (fever) and biological predictors, such as ESR, serum WBC count, C-reactive protein (CRP), and immunoglobulin G and antinuclear antibodies [18].

In pediatric populations with mean ages between 5.3 and 6.1 years old, Kocher et al. (1999) retrospectively distinguished three patient groups: (1) true SAH, based on a JFA WBC count ≥50,000/mm^3^ and positive blood cultures or a positive JFA culture; (2) presumed SAH, based on a JFA WBC count ≥50,000/mm^3^ and negative blood and JFA cultures; and 3) true TSH, based on a JFA WBC count <50,000/mm^3^ and negative blood and JFA cultures [15]. The groups’ clinical and biological variables were compared using univariate analyses and multiple logistic regressions. Predictors were defined, compared, and combined between the groups, and their diagnostic accuracy was also evaluated. Kocher et al.’s analysis determined four independent multivariate predictors: a history of fever, non-weight bearing, an ESR ≥ 40 mm/h, and a serum WBC count > 12,000/mm^3^. Their combination resulted in a probability of SAH rather than TSH of <0.2% for zero predictors, 3% for one predictor, 40% for two, 93.1% for three, and 99.6% for four [15].

This clinical prediction rule was then validated by repeating the statistical process in a prospective cohort of patients with SAH and TSH, and it successfully differentiated the three groups of patients (true SAH, presumed SAH and true TSH) [30]. The four independent multivariate predictors were confirmed, although with different results for the probability of SAH obtained when combining the factors. The probabilities were higher than in the retrospective study for zero or one predictor and lower for two–four predictors (2% for zero predictors, 9.5% for one predictor, 35% for two, 72.8% for three, and 93% for four), with a smaller area under the receiver operating characteristic curve of 0.86 vs. 0.96 in the first study. The authors proposed no explanations for these differences. By appraising Kocher’s papers, they could perhaps be attributed to the populations’ different clinical and biological characteristics, to retrospective rather than prospective data collection, or to the different sample sizes (82 SAH and 86 TSH patients in the 1999 paper; 51 SAH and 103 TSH patients in the 2004 paper) [15,30].

It is interesting to note that the prospective validation study’s SAH group included 51 patients, of whom 24 were culture-positive and 27 culture-negative. At this point, the Kocher criteria were used to try to find common biological markers among SAH patients who were probably infected by different pathogens. Indeed, Kocher suggested that “organisms that were difficult to grow on culture, viral arthritis, arthritis resulting from atypical organisms” could have affected patients in the presumed SAH group, and this was in the decade when *Kingella kingae*-related pediatric osteoarticular infections were just emerging [30]. Moreover, the clinical implications proposed by Kocher were remarkable: close clinical observations in the case of zero predictors (which is especially relevant for culture-negative patients) and JFA in the operating room followed by arthrotomy for patients with four predictors. These suggestions were futuristic at a time when *K. kingae* was just starting to emerge [30].

Among the 24 culture-positive samples in Kocher’s prospective study, 16 patients had positive blood cultures and JFA cultures, 6 only had a positive JFA culture, and 2 only had a positive blood culture. Gram staining on JFAs was only positive for 17 of the 22 JFA culture-positive patients. This is quite interesting since it is well known that the time to the first results of blood cultures or JFA cultures can be more than 48 h, and, in this timeframe, patients undergo arthrotomy, wash-out, and empiric intravenous antibiotic therapy based on the clinical evaluation of their JFA and WBC count. Moreover, among the patients excluded, five (four treated with arthrotomy and one with antibiotics) were considered and treated as false negatives based on their negative cultures and low JFA WBC count. The Koche criteria’s worse performance in the prospective study than in the retrospective one further reinforced the need to obtain reliable biomarkers enabling a robust differential diagnosis between SAH and TSH before an arthrotomy or any other therapeutic strategy. 

One challenge for clinical practice and research in the 20th century and into the 2010s was the availability of a standardized unit of measurement for serum biomarker results. This was the case for CRP, which appeared patchily in the literature as of the 1980s, with studies initially focusing on CRP trends in the days following infection and surgery [18,32,34,39,40,41]. Kunnamo et al. were among the first to study CRP when searching for predictors with which to differentiate between pediatric SA of any joint and juvenile arthritis [18].

In 2006, Caird et al. included the CRP in their prospective validation study of Kocher’s criteria, adopting 20 mg/L as a threshold, as did the studies by Kallio et al. and Eich et al. [13,32,42]. Even though they used Kocher’s classes to distinguish between true and presumed SAH and TSH, the SAH group included four patients with a JFA WBC count < 50,000/mm^3^, two culture-negative patients treated conservatively with antibiotics, and one culture-negative patient treated surgically. Their overall sample size was smaller, with 34 patients in the SAH group and 14 in the TSH group. A fever > 38.5 °C was only recorded in patients with SAH. A univariate analysis, multiple logistic regression, and Akaike’s information criterion were used to validate the independent predictors. The individual predictors of SAH were non-weight bearing, ESR ≥ 40 mm/h, a WBC count > 12,000/mm^3^, and a CRP > 20 mg/L. Their combination provided a probability of SAH rather than TSH of 16.9% for zero predictors, 36.7% for one predictor, 62.4% for two, 82.6% for three, 93.1% for four, and 97.5% for five. Despite the probability of SAH being high for of a combination of at least three criteria, it must be noted that for zero criteria, the probability was 16.9%, which could call into question the accuracy of these predictors, as they were not studied with sensitivity or specificity tests, as Kocher had done. 

## 4. Validation and Limitations of Kocher–Caird Prediction Rules

Multiple authors have replicated the validation of Kocher–Caird predictors in different geographical settings, not only for SAH but also for other joints [43,44,45]. However, they sometimes used different criteria to a priori define SAH versus a non-septic arthritis pathology [31,33,34,43,46,47,48]. Due to these methodological differences, the Kocher–Caird criteria were not validated in these particular cases [31,33,34,43,46,47,48]. 

Apart from the predictors, it was evident in both Kocher’s and Caird’s studies that the JFA WBC count threshold of 50,000/mm^3^—upon which the classification of true and presumed SAH and TSH and their entire methodology was based—was still the source of uncertainty for the diagnosis and consequent management of SAH. Using a retrospective prognostic case series, Kocher’s team studied the JFA WBC count parameter further to better differentiate between SAH, Lyme’s disease, TSH, and other orthopedic conditions (described as inflammatory or rheumatological) in patients with an intermediate JFA WBC count of 25,000 to 75,000/mm^3^ by broadening the inclusion criteria to all patients aged <19 years old [16]. This retrospective study allocated 46 patients with diagnostic certitude to the aforementioned groups, 15 of whom were in the SAH group. The overall sample was further distinguished between low (<50,000/mm^3^) and high (>50,000/mm^3^) JFA WBC counts. Twenty-three patients were allocated to each of the high- and low-JFA-WBC-count groups, with 4/23 (17%) SAH and 6/23 (26%) TSH patients in the low-count group and 11/23 (48%) SAH and 2/23 (9%) TSH patients in the high-count group. As one might have hypothesized, the statistical analysis concluded that an SAH patient was 4.4 times more likely to have a high JFA WBC count than a low one. Nevertheless, the authors noted that three of the four SAH patients with a low JFA WBC count were culture-positive, and the combination of Kocher–Caird predictors did not modify the odds of these patients having SAH. The authors concluded, therefore, that even in cases involving a low JFA WBC count, SAH “should be high on the differential diagnosis” [16]. Such a result could be due not only to the small sample size but also to the distribution and frequency of SAH and TSH in both groups. Despite this long-standing debate, the question of a suitable JFA WBC count threshold was finally answered during the last decade thanks to a large case series [49,50]. Currently, the accepted JFA WBC count threshold for pediatric patients with septic arthritis is 50,000/mm^3^ as per the Pediatric Infectious Diseases Society and the Infectious Diseases Society of America [49,50,51].

## 5. Distinguishing Septic Arthritis of the Hip Related to Pyogenic Bacteria versus *Kingella kingae*

The main pathogens found in the studies by Kocher and Caird were *Staphylococcus aureus*, *Streptococcus pneumoniae*, *Haemophilus influenzae*, *Neisseria meningitidis*, and *Streptococcus pyogenes* [15,30,32]. In the decade when Kocher’s first work was published, *K. kingae* was emerging onto the global stage in the world of pediatric osteoarticular infection [52,53,54,55]. One of the challenges in its microbiological identification was the difficulty isolating it when the JFA or blood cultures were cultured in routine solid media, such as blood–agar or chocolate–agar plates, identifying *K. kingae* as an additional trypticase soy agar to standard blood–agar, chocolate–agar or 2 micrograms/mL of vancomycin to inhibit the competing bacterial flora [56]. The advent of nucleic acid amplification assays (NAAAs) improved *K. kingae* detection, making it possible even in cases when patients have undergone prior treatment with antibiotics, with a shorter delay [57]. Nevertheless, NAAA use was affected by the time required for its spread worldwide. Pediatric osteoarticular infections (OAIs) related to *K. kingae* have a mild presentation, with moderate or low local and systemic signs of inflammation, and their diagnosis thus requires a high clinical suspicion [58]. Various studies have found that *K. kingae* was the pathogen responsible for 30–93% of pediatric OAIs, including in the spine, intervertebral discs, and tendon sheaths, among children aged 6–48 months [56,58,59,60,61,62]. Its epidemiology diminished the role of *S. aureus*, which had been considered the predominant pathogen responsible for pediatric OAIs for many decades [54,56,63,64,65,66,67,68,69,70].

Because it is a pathogen that colonizes the oropharynx, *K. kingae* can disseminate into the bloodstream, provoking a hematogenous OAI [71]. For this reason, testing oropharyngeal swabs can be a rapid means of diagnosing *K. kingae*-related OAIs in children aged 6–48 months old [56,72,73]. However, oropharyngeal colonization with *K. kingae* does not automatically translate into a *K. kingae*-related OAI. The colonization of children aged 6–48 months old by *K. kingae* has been suggested by studies showing a very low prevalence (0.67%) of positive oropharyngeal swabs in asymptomatic babies younger than 6 months old, as well as the colonization of adults living with children aged 6–48 months [74,75]. Moreover, a case–control study by Gravel et al. (2017) in infants 6–48 months old showed that 6% of controls had a positive *K. kingae* oropharyngeal swab in the absence of signs or symptoms of an OAI [76]. Indeed, in this study, the odds ratio for an association between a positive oropharyngeal swab and an OAI was 38.3 [76]. *K. kingae* was nevertheless isolated in the cultures of 3/19 patients with a negative oropharyngeal swab [76].

When differentiating *K. kingae* infections from Gram-positive coccal infections, the former are characterized by a temperature < 38 °C at admission, a serum CRP < 55 mg/L, a WBC count < 14,000/mm^3^, and band forms < 150/mm^3^ [77]. Similarly, when differentiating *K. kingae* infections from methicillin-sensitive *S. aureus* OAIs, the former are characterized by patients aged < 43 months, a temperature < 37.9 °C, a CRP < 32.5 mg/L, and a platelet count > 361,500/mm^3^ at admission [78]. It is of note that in the study comparing *K. kingae* and methicillin-sensitive *S. aureus*, the criteria with the highest negative predictive values for a *K. kingae* infection were an age ≥ 43 months and a CRP ≥ 32.5 mg/L (at 96% and 80%, respectively) [78].

Once the features of a *K. kingae* OAI had been defined, there arose the suspicion that *K. kingae* might be the fastidious pathogen in Kocher–Caird culture-negative patients because of the similarities with the clinical and biological profiles of these patients [2,56,79,80,81]. An Israeli–Spanish *K. kingae* research group thus studied Kocher’s criteria [15] in a population of *K. kingae*-related SAH patients [35]. This involved a small sample size of 34 children, without a comparative group of TSH patients, and it also applied a statistical analysis different from Kocher’s, thus limiting a strict comparison between these studies. A higher sensitivity (88%) was only achieved for the refusal-to-bear-weight criterion. Seventy-one % of patients with *K. kingae*-related SAH were positive to ≤ two Kocher criteria, and therefore had a ≤40% probability of SAH [35]. More recently, Hagedoorn et al. stratified a sample of 129 preschool SAH patients (between 3 months and 5 years old) into four groups determined by their pathogens (*S. aureus, K. kingae,* other pathogens, and culture-negative patients) to validate the Kocher–Caird criteria [37]. The sensitivity of the *S. aureus* and *K. kingae* groups was similar (about 60%), but the overall sensitivity of the Kocher–Caird criteria was 56.6%, suggesting the need for new clinical prediction rules for pediatric SAH.

## 6. New Perspectives for Future Prediction Rules for Pediatric Septic Arthritis of the Hip

By contextualizing the Kocher–Caird criteria and their use in the *Kingella* era, the following points must be taken into consideration. The criteria were originally validated in studies comparing SAH and TSH in children with a wide range of ages rather than covering TSH’s primary presumed epidemiological age range of between 3 and 10 years old [15,32,39]. No age range subgroups have been provided in Kocher–Caird studies [15,32,39]. It cannot be assumed or demonstrated, therefore, that the culture-negative group really fits the *K. kingae* profile. Moreover, the setting where the criteria were individualized and validated was in a priori-determined classes of patients—true SAH, presumed SAH, and true TSH—based on a JFA WBC count threshold of 50,000/mm^3^ and the results of JFA and blood cultures. As discussed, the criteria for the three classes of patients troubled Kocher and Caird, obliging them to either exclude or include a few patients from or in classes in which they should have been a priori categorized. Interestingly, years later, it has been shown that conditions like juvenile idiopathic arthritis can have JFA WBC counts of > 50,000/mm^3^, which makes them wrongly eligible for the class of presumed SAH adopted by Kocher and Caird [15,30,32,49]. The latter result carries the risk that the Kocher–Caird criteria could be misleading, owing to the degrees of diagnostic uncertainty accepted within the definitions of the three groups of true SAH, presumed SAH, and true TSH. 

Considering all this, due to the risk of misdiagnosing SAH, TSH should only be considered using a process of exclusion diagnosis. Further studies are required to define the biological markers that can unquestionably distinguish pediatric SAH (especially when due to *K. kingae*) and its mimics. To do this, it is also crucial to take into consideration that the clinical and biological aspects of SAH are closely related to a child’s age and, above all, the pathogen responsible. According to the research on predictive criteria for *K. kingae* OAIs, they are most reliable for children aged 6–48 months old [58,77,78]. Biological markers still require further investigation in large samples. Among them, CRP seems to be promising, with a threshold varying from 32.5 to 55 mg/mL [77,78]. It is also important that the previously proposed criteria for *K. kingae* OAIs [77,78] are studied in multiple countries. The methodology for the validation of prediction rules requires using retrospective cases series first to find reliable criteria in multiple settings that can then be further tested in prospective studies. Subsequently, using a similar methodology, the criteria should be tested for their reliability to distinguish between *Kingella*-related and pyogenic-related infections. A solid methodology is essential for discovering reliable criteria for a prompt diagnosis of SAH in children. Distinguishing the class of pathogen (*K. kingae* vs. pyogens) is essential, since the therapeutic strategies for dealing with them are very different. OAIs due to *K. kingae* are characterized by weaker general and local inflammatory reactions and no long-term orthopedic sequelae, making them treatable using antibiotic therapy alone, without surgery [59].

A robust methodology is the basis for the advancing research on pediatric OAIs. Further collaboration and networking between centers and clinicians can contribute significantly to improving the management of pediatric OAI worldwide.

## Figures and Tables

**Table 1 microorganisms-12-00550-t001:** Studies on the diagnostic criteria for septic arthritis and transient synovitis of the hip.

Author and Year	Study Design	Comparison Groups	Age	Criteria Suggested for Differential Diagnosis
Edwards, 1951 [10]	Retrospective case series and narrative review	TSH—no comparison	Range: 4–6 y	sWBC normal to 12,000/mm^3^. ESR elevation. Absence of bacteria in the JFA.
Rosenberg et al., 1956 [19]	Retrospective case series	TSH—no comparison	Range: 2.5–12 y	BT < 100 °F.
Hardinge, 1970 [20]	Case–control	TSH vs. controls	Range: 3–10 y	sWBC normal (threshold age group-dependent). No growth of blood cultures. ESR and sWBC curves. Throat swabs, antristreptolysin. Chest radiograph, urine analysis, Mantoux, latex fixation for RA.
Marchal et al., 1987 [21]	Retrospective case series	Irritable hip (TSH, SAH, other etiologies)	Range: 10 m–17 y	ESR, sWBC, neutrophils count, CRP, antistreptolysin. US.
Lohmander et al., 1988 [22]	Retrospective case series	Irritable hip (TSH, SAH, other etiologies)	Range: 4 m–15 y	Previous episode, duration of symptoms. BT. ESR. JFA proteoglycan antigen
Bennett et al., 1992 [23]	Retrospective case series	SAH	Range: 3–11 y	BT > 38 °C, pain to the hip reproducible by passive motion. JFA with pus, fWBC > 50,000 mm^3^, JFA Gram stain or culture. RX
Del Beccaro et al., 1992 [24]	Retrospective case series	TSH vs. SAH	Range: 2.5 w–17 y	BT > 37.5 °C, ESR > 20 mm/h: their combination provides 97% SAH. sWBC, neutrophil count.
Chen et al., 1993 [25]	Retrospective case series	SAH	n.a. (article in chinese)	BT, sWBC, ESR, blood cultures, JFA cultures.
Zawin et al., 1993 [14]	Retrospective case series	Irritable hip (TSH, SAH, other etiologies)	Range: 6 w–15 y	Clinical evaluation, RX, US, sWBC. Significant difference between TSH and SAH: ESR, jWBC > 20,000 mm^3^, JFA Gram stain.
Taylor and Clarke, 1994 [26]	Retrospective case series	Irritable hip (TSH, SAH, other etiologies)	Mean: 5.4 y	Significant parameters: severe spasm, tenderness, BC ≥ 38 °C, ESR ≥ 20 mm/h (the combination of any two: specificity 91% and sensitivity 95% for sepsis). Neutrophil count: not significative.
Fink et al., 1995 [27]	Retrospective case series	Irritable hip (TSH, SAH, other etiologies)	Range: 1–10 y	US, JFA stain, bone scintigraphy.
Klein et al., 1997 [28]	Retrospective case series	SAH	Range: 0–6 y	BT, sWBC, ESR. Most sensitive parameter: ESR.
Eich et al., 1999 [13]	Retrospective case series	Irritable hip (TSH, SAH, other etiologies)	Range: 1 m–12.5 y	US, rectal BT ≥ 38 °C, ESR ≥ 10 mm/h, CRP ≥ 10 mg/L, sWBC (different thresholds according to the age of patients).
Kocher et al., 1999 [15]	Retrospective case series	SAH (true and presumed) vs. TSH	Mean (for SAH): 6 ± 4.2 y	BT ≥ 38.5 °C, NWB, ESR > 40 mm/h, sWBC > 12,000 mm/^3^. Combination of four predictors: 99.6% probability of SAH.
Jung et al., 2003 [29]	Retrospective case series	SAH vs. TSH	Range: 1 m–15 y	Predictors of SAH with a probability of 98.6%: BT > 37 °C, ESR > 20 mm/h, CRP > 1 mg/dL, sWBC > 11,000/mL, RX showing increased hip joint space of >2 mm.
Kocher et al., 2004 [30]	Prospective cohorts	SAH (true and presumed) vs. TSH	Mean (for SAH): 5.7 ± 3.6 y	BT ≥ 38.5 °C, NWB, ESR > 40 mm/h, sWBC > 12,000 mm/^3^. Combination of four predictors: 86% probability of SAH.
Luhmann et al., 2004 [31]	Retrospective diagnostic study	SAH (true and presumed) vs. TSH	Mean (for SAH): 63.4 ± 45.7 m	BT ≥ 38.5 °C, NWB, ESR > 40 mm/h, sWBC > 12,000 mm/^3^. Combination of four predictors: 59% probability of SAH. Combination of BT, sWBC > 12,000 mm/^3^, previous healthcare visit: 71% probability of SAH.
Caird et al., 2006 [32]	Prospective cohorts	SAH (true and presumed) vs. TSH	Range: 7 m–16 y. Mean 5.5 y	Oral BT > 38.5 °C, CRP > 20 mg/L, NWB, ESR > 40 mm/h, sWBC > 12,000 mm/^3^. Combination of five predictors: 97.5% probability of SAH.
Sultan et al., 2010 [33]	Retrospective diagnostic study	SAH vs. TSH	Range: 1–12 y	BT ≥ 38.5 °C, CRP > 20 mg/L, NWB, ESR > 40 mm/h, sWBC > 12,000 mm/^3^. Combination of five predictors: 59.9% probability of SAH.
Singhal et al., 2011 [34]	Retrospective diagnostic study	SAH vs. TSH	Mean age: 5.3 y	BT ≥ 38.5 °C, CRP > 20 mg/L, NWB, ESR > 40 mm/h, sWBC > 12,000 mm/^3^. NWB and CRP > 20 mg/L: 74% probability of SAH.
Yagupsky et al., 2014 [35]	Retrospective diagnostic study	SAH vs. TSH	Range: 6–27 m	BT ≥ 38.5 °C, NWB, ESR > 40 mm/h, sWBC > 12,000 mm/^3^. Most Kingella patients (71%) have ≤2 Kocher criteria.
Clever et al., 2021 [36]	Retrospective diagnostic study	Irritable hip (TSH, SAH, other etiologies)	Mean (for SAH): 67.1 ± 35.4 m	CRP, ESR, sWBC described. Transforming growth factor alpha, IL-7, IL-33, IL-28a: sensitivity and specificity 90.9%.
Hagedoorn et al., 2023 [37]	Retrospective diagnostic study	SAH	Median: 19 m	Oral BT > 38.5 °C, CRP > 20 mg/L, NWB, ESR > 40 mm/h, sWBC > 12,000 mm/^3^. Tested in patients infected by *Kingella kingae* and *Staphylococcus aureus*. Combination of four and five predictors: 56.6% probability of SAH.
Olandres et al., 2023 [38]	Retrospective diagnostic study	SAH vs. TSH	Median: 8 y	Oral BT > 38.5 °C, CRP > 20 mg/L, NWB, ESR > 40 mm/h, sWBC > 12,000 mm/^3^. Combination of four predictors: 59.16% specificity for SAH and poor sensitivity (0%). CRP ≥ 20 mg/L and US (effusion ≥ 7 mm): specificity 97%, sensitivity 71% for SAH.

Legenda: BT: body temperature; CRP: C-reactive protein; ESR: erythrocyte sedimentation rate; JFA: joint fluid aspirate; m: months; n.a.: not available; NWB: non weight bearing; RA: rheumatoid arthritis; RX: radiography; SAH: septic arthritis of the hip; jWBC: joint fluid white blood cell count; sWBC: serum white blood cell count; TSH: transient synovitis of the hip; US: ultrasound imaging; y: years; w: weeks.

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
