# Peer review of "The Kocher–Caird Criteria for Pediatric Septic Arthritis of the Hip: Time for a Change in the Kingella Era?"

_microorganisms, 2024, doi:10.3390/microorganisms12030550_

Round 1
Reviewer 1 Report
Comments and Suggestions for Authors
This manuscript needs more focus on the Kocher–Caird criteria and their predictive value in detecting SAH of the hip due to pathogens other than Kingella and due to Kingella, with inclusion of tables to show these points.
Discussion of culture methods to optimize recovery of Kingella would be very useful
Points by line number:
9-12 The abstract provides zero information. It should include details of the Kocher–Caird criteria and their predictive value in detecting SAH of the hip due to pathogens other than Kingella and due to Kingella.
109 Area under what curve?
117-122 This long sentence does not make sense
130 and 133 Use blood culture not hemoculture
138-140 This long sentence needs editing
160 Serum is not needed
191 underlined should be noted or emphasized
202 Pyogenic is not the correct term here – pyogenic usually refers to staphylococcal and some streptococcal infections
205 Should be Haemophilus
205-106 group A Streptococcus should be Streptococcus pyogenes
209-210 Routine culture media include blood agar and, if appropriate, chocolate agar. Kingella often does not grow despite use of these media.
219 Downsized should be diminished
221-224 Has the correlation between colonization with Kingella and infection been established? Similar unfounded claims have been made about colonization with S. pneumoniae and H. influenzae, which have poor positive predictive value in determining infection (but better negative predictive value in excluding infection)
235 cocci should be coccal
271 individuated should be individualized
283-292 The concluding paragraph is very disappointing – it basically states that existing criteria doe not reliably distinguish between SAH and TSH and that further studies are needed.
Comments on the Quality of English LanguageSome editing of terminology and long sentences would be helpful as noted in my review.
Author Response
Dear Reviewer,
Thank you for your constructive feedback.
We have corrected the paper as you suggested. The details are provided below.
We have added a table on the studies on transient synovitis and septic arthritis of the hip.
We look forward to your feedback.
The Authors
Details:
Discussion of culture methods to optimize recovery of Kingella would be very useful – from line 242.
9-23, abstract: We developed the abstract focusing on the history of the Kocher–Caird criteria and on the rationale for our historical review. We decided to leave the discussion on the predictive criteria for Kingella in the manuscript rather than have it in the abstract, for reasons of abstract length.
109 --> 123: receiver operating characteristic curve
117-122 --> from 133: sentence reformulated
130 and 133 --> 144 and 145: blood culture (we replaced hemoculture with blood culture throughout the paper)
138-140 --> from 152: sentence reformulated
160 --> 188: serum deleted
191 --> 218: replaced
202 --> 235: we left pyogenic since most of the non-Kingella pathogens are Staphylococci or Streptococci
205 --> 238: Haemophilus
205-6 --> 238: Streptococcus pyogenes
209-210 --> from 242: rectified
219 --> 254: diminished
221-224 --> explained from line 259
235 --> 270: coccal
271 --> 314: individualized
283-292 --> from 326: this review provides another point of view on the challenges of diagnosing pediatric septic arthritis of the hip, that is the historical excursus, and differentiating Kingella kingae. We do not have the solution for the diagnostic criteria question, but the manuscript suggests directions for future research to investigate.
Reviewer 2 Report
Comments and Suggestions for Authors
The manuscript titled "The Kocher–Caird criteria for pediatric septic arthritis of the hip: time for a change in the Kingella era?" by Valisena Silvia et al. provides an extensive review of the clinical and laboratory test algorithms for diagnosing pediatric septic arthritis of the hip (SAH) and proposes new research directions for improving prediction rules for SAH in children, especially in light of emerging pathogens such as Kingella kingae.
Strengths:
- Comprehensive Review: The manuscript offers a thorough historical overview of diagnostic approaches for SAH, including the evolution of clinical prediction rules and the impact of emerging pathogens.
- Critical Analysis: The authors critically assess the limitations of current diagnostic criteria, particularly the Kocher–Caird criteria, in the context of modern microbiological advances.
- Future Directions: It proposes valuable research directions to refine diagnostic algorithms and incorporate new biomarkers, which could lead to more accurate and timely diagnoses of SAH.
Areas for Improvement:
- Methodological Details: The manuscript could benefit from more detailed descriptions of the proposed methodologies for future research, including specific biomarkers to be investigated and the design of prospective studies.
- Statistical Analysis: While the manuscript discusses previous studies' findings, it could further elaborate on the statistical methods used in those studies and how future research might address their limitations.
- Clinical Implications: The discussion on the clinical implications of adopting new diagnostic criteria or biomarkers could be expanded, providing a clearer picture of how these changes might influence pediatric orthopedic practice.
Conclusion:
The manuscript by Valisena Silvia et al. contributes significantly to the ongoing discourse on improving diagnostic criteria for pediatric SAH. It highlights the need for research that takes into account the changing epidemiology of causative pathogens. Further work to refine and validate new diagnostic criteria will be crucial for enhancing patient outcomes in pediatric orthopedics.
Author Response
Dear Reviewer,
Thank you for your constructive feedback.
We have modified the paper as you suggested. The details are provided below.
We have added a table on the studies on transient synovitis and septic arthritis of the hip.
We look forward to your feedback.
The Authors
Details:
Areas for improvement:
- Methodological details: developed in section 6
- Statistical analysis: we thought about an appraisal of the statistical analyses during our review. However, when we went to the bibliographic references for the statistics of the studies by Kocher et al., we realized that multiple statistical models had been used for this research and that it had evolved over the years. Comparing the statistical bibliography and methods with a letter by Uzoigwe (published in CORR) that appraised Kocher et al.’s statistics, we found that some criticisms of their statistical model were probably not completely correct. Such a historical statistical appraisal was challenging because of the differences between statistical models, which would require strong competencies in statistics to fathom. Since the methodologies of the studies appraised provided good material for the manuscript and the statistical challenges were far beyond our competencies, we decided to defer any statistical excursus.
- Clinical implications: developed in section 6.